# INTERPRETING CLIP'S IMAGE REPRESENTATION VIA TEXT-BASED DECOMPOSITION

**Yossi Gandelsman, Alexei A. Efros, and Jacob Steinhardt**
UC Berkeley
{yossi_gandelsman,aaefros,jsteinhardt}@berkeley.edu

## ABSTRACT

We investigate the CLIP image encoder by analyzing how individual model components affect the final representation. We decompose the image representation as a sum across individual image patches, model layers, and attention heads, and use CLIP's text representation to interpret the summands. Interpreting the attention heads, we characterize each head's role by automatically finding text representations that span its output space, which reveals property-specific roles for many heads (e.g. location or shape). Next, interpreting the image patches, we uncover an emergent spatial localization within CLIP. Finally, we use this understanding to remove spurious features from CLIP and to create a strong zero-shot image segmenter. Our results indicate that a scalable understanding of transformer models is attainable and can be used to repair and improve models.[1]

## 1 INTRODUCTION

Recently, Radford et al. (2021) introduced CLIP, a class of neural networks that produce image representations from natural language supervision. As language is more expressive than previously used supervision signals (e.g. object categories) and CLIP is trained on a lot more data, its representations have proved useful on downstream tasks including classification (Zhou et al., 2022), segmentation (Lüddecke & Ecker, 2022), and generation (Rombach et al., 2022). However, we have a limited understanding of what information is actually encoded in these representations.

To better understand CLIP, we design methods to study its internal structure, focusing on CLIP-ViT (Dosovitskiy et al., 2021). Our methods leverage several aspects of CLIP-ViT's architecture: First, the architecture uses *residual* connections, so the output is a sum of individual layer outputs. Moreover, it uses *attention*, so the output is also a sum across individual locations in the image. Finally, the representation lives in a joint vision-language space, so we can label its directions with text. We use these properties to decompose the representation into text-explainable directions that are attributed to specific attention heads and image locations.

As a preliminary step, we use the residual structure to investigate which layers have a significant direct effect on the output. We find that ablating all layers but the last 4 attention layers has only a small effect on CLIP's zero-shot classification accuracy (Section 3). We conclude that the CLIP image representation is primarily constructed by these late attention layers.

We next investigate the late attention layers in detail, leveraging the language space to uncover interpretable structure. We propose an algorithm, TEXTSPAN, that finds a basis for each attention head where each basis vector is labeled by a text description. The resulting bases reveal specialized roles for each head: for example, one head's top 3 basis directions are *A semicircular arch*, *A isosceles triangle* and *oval*, suggesting that it specializes in shapes (Figure 1(a)).

We present two applications of these identified head roles. First, we can reduce spurious correlations by removing heads associated with the spurious cue; we apply this on the Waterbirds dataset (Sagawa et al., 2019) to improve worst-group accuracy from 48% to 73%. Second, the representations of heads with a property-specific role can be used to retrieve images according to that property; we use it to perform retrieval based on discovered senses of similarity, such as color, location, and texture.

We next exploit the spatial structure provided by attention layers. Each attention head's output is a weighted sum across image locations, allowing us to decompose the output across these locations.

---

[1]Project page and code: https://yossigandelsman.github.io/clip_decomposition/

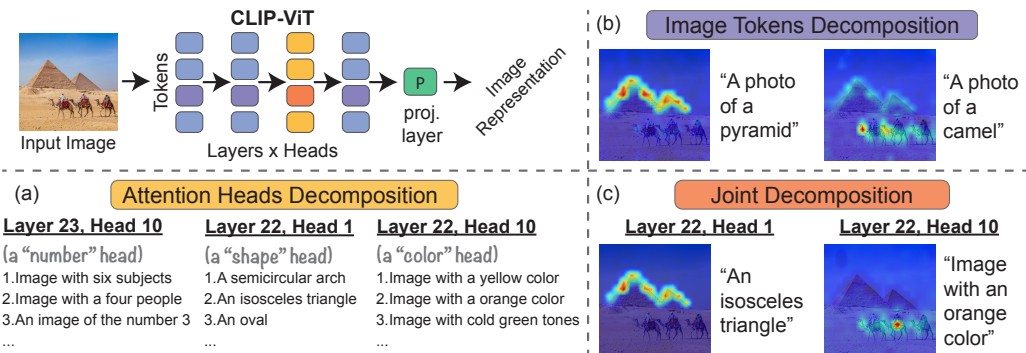

Figure 1: **CLIP-ViT image representation decomposition.** By decomposing CLIP's image representation as a sum across individual image patches, model layers, and attention heads, we can (a) characterize each head's role by automatically finding text-interpretable directions that span its output space, (b) highlight the image regions that contribute to the similarity score between image and text, and (c) present what regions contribute towards a found text direction at a specific head.

We use this to visualize how much each location writes along a given text direction (Figure 1(b)). This yields a zero-shot image segmenter that outperforms existing CLIP-based zero-shot methods.

Finally, we consider the spatial structure jointly with the text basis obtained from TEXTSPAN. For each direction in the basis, the spatial decomposition highlights which image regions affect that basis direction. We visualize this in Figure 1(c), and find that it validates our text labels: for instance, the regions with triangles are the primary contributors to a direction that is labeled as *isosceles triangle*.

In summary, we interpret CLIP's image representation by decomposing it into text-interpretable elements that are attributed to individual attention heads and image locations. We discover property-specific heads and emergent localization, and use our discoveries to reduce spurious cues and improve zero-shot segmentation, showing that understanding can improve downstream performance.

## 2 RELATED WORK

**Vision model explainability.** A widely used class of explainability methods produces heatmaps to highlight parts in the input image that are most significant to the model output (Selvaraju et al., 2017; Sundararajan et al., 2017; Binder et al., 2016; Voita et al., 2019; Lundberg & Lee, 2017; Chefer et al., 2021). While these heatmaps are useful for explaining the relevance of specific image regions to the output, they do not show how attributes that lack spatial localization (e.g. object size or shape) affect the output. To address this, a few methods interpret models by finding counterfactual edits using generative models (Goetschalckx et al., 2019; Lang et al., 2021; Aberman et al., 2021). All these methods aim to explain the output of the model without interpreting its intermediate computation.

**Intermediate representations interpretability.** An alternate way to explain vision models is to study their inner workings. One approach is to invert intermediate features into the input image space (Dosovitskiy & Brox, 2015; Mahendran & Vedaldi, 2014; Goh et al., 2021). Another approach is to interpret individual neurons (Bau et al., 2020; 2019; Dravid et al., 2023) and connections between neurons (Olah et al., 2020). These approaches interpret models by relying only on visual outputs.

Few methods use text to interpret intermediate representations in vision models. Hernandez et al. (2022) provide text descriptions for image regions in which a neuron is active. Yuksekgonul et al. (2023) project model features into a bank of text-based concepts. More closely to us, a few methods analyze CLIP's intermediate representations via text—Goh et al. (2021) find multimodal neurons in CLIP that respond to different renditions of the same subject in images. Materzynska et al. (2022) study entanglement in CLIP between images of words and natural images. We differ from these works by using CLIP's intrinsic language-image space and by exploiting decompositions in CLIP's architecture for interpreting intermediate representations.

**Contrastive vision-language models.** Contrastive vision-and-language models like CLIP (Radford et al., 2021) and ALIGN (Jia et al., 2021) showed promising zero-shot transfer capabilities for downstream tasks, including OCR, geo-localization and classification (Wortsman, 2023). Moreover, CLIP

representations are used for segmentation (Lüddecke & Ecker, 2022), querying 3D scenes (Kerr et al., 2023), and text-based image generation (Ramesh et al., 2021; Rombach et al., 2022). We aim to interpret what information is encoded in these representations.

# 3    DECOMPOSING CLIP IMAGE REPRESENTATION INTO LAYERS

We start by presenting the CLIP model (Radford et al., 2021) and describe how the image representation of CLIP-ViT is computed. We show that this representation can be decomposed into direct contributions of individual layers of the image encoder ViT architecture. Through this decomposition, we find that the last few attention layers have most of the direct effects on this representation.

## 3.1    CLIP-ViT PRELIMINARIES

**Contrastive pre-training.** CLIP is trained to produce visual representations from images $I$ coupled with text descriptions $t$. It uses two encoders—a transformer-based text encoder $M_{\text{text}}$ and an image encoder $M_{\text{image}}$. Both $M_{\text{text}}$ and $M_{\text{image}}$ map to a shared vision-and-language latent space, allowing us to measure similarity between images and text via cosine similarity:

$$\text{sim}(I,t) = \langle M_{\text{image}}(I), M_{\text{text}}(t) \rangle / (||M_{\text{image}}(I)||_2 ||M_{\text{text}}(t)||_2) \tag{1}$$

Given a batch of images and corresponding text descriptions $\{(I_i, t_i)\}_{i \in \{1,...,k\}}$, CLIP is trained to maximize the similarity of the image representation $M_{\text{image}}(I_i)$ to its corresponding text representation $M_{\text{text}}(t_i)$, while minimizing $\text{sim}(I_i, t_j)$ for every $i \neq j$ in the batch.

**Zero-shot classification.** CLIP can be used for zero-shot image classification. To classify an image given a fixed set of classes, each name of a class (e.g. "Chihuahua") is mapped to a fixed template (e.g. "An image of a {class}") and encoded by the CLIP text encoder. The prediction for a given image is the class whose text description has the highest similarity to the image representation.

**CLIP image representation.** Several architectures have been proposed for computing CLIP's image representation. We focus on the variant that incorporates ViT (Dosovitskiy et al., 2021) as a backbone. Here a vision transformer (ViT) is applied to the input image $I \in \mathbb{R}^{H \times W \times 3}$ to obtain a $d$-dimensional representation $\text{ViT}(I)$. The CLIP image representation $M_{\text{image}}(I)$ is a linear projection of this output to a $d'$-dimensional representation in the joint vision-and-language space[2]. Formally, denoting the projection matrix by $P \in \mathbb{R}^{d' \times d}$:

$$M_{\text{image}}(I) = P\text{ViT}(I) \tag{2}$$

Both the parameters of the ViT and the projection matrix $P$ are learned during training.

**ViT architecture.** ViT is a residual network built from $L$ layers, each of which contains a multi-head self-attention (MSA) followed by an MLP block. The input $I$ is first split into $N$ non-overlapping image patches. The patches are projected linearly into $N$ $d$-dimensional vectors, and positional embeddings are added to them to create the *image tokens* $\{z_i^0\}_{i \in \{1,...,N\}}$. An additional learned token $z_0^0 \in \mathbb{R}^d$, named the *class token*, is also included and later used as the output token.

Formally, the matrix $Z^0 \in \mathbb{R}^{d \times (N+1)}$, with the tokens $z_0^0, z_1^0, ..., z_N^0$ as columns, constitutes the initial state of the residual stream. It is updated for $L$ iterations via these two residual steps:

$$\hat{Z}^l = \text{MSA}^l(Z^{l-1}) + Z^{l-1}, \quad Z^l = \text{MLP}^l(\hat{Z}^l) + \hat{Z}^l. \tag{3}$$

We denote the first column in the residual stream $Z^l$, corresponding to the class token, by $[Z^l]_{cls}$. The output of the ViT is therefore $[Z^L]_{cls}$.

## 3.2    DECOMPOSITION INTO LAYERS

The residual structure of ViT allows us to express its output as a sum of the direct contributions of individual layers of the model. Recall that the image representation $M_{\text{image}}(I)$ is a linear projection

---

[2]Both here and in Eq. 3, we ignore a layer-normalization term to simplify derivations. We address layer-normalization in detail in Section A.1.

| | Base accuracy | + MLPs ablation |
|---|---|---|
| ViT-B-16 | 70.22 | 67.04 |
| ViT-L-14 | 75.25 | 74.12 |
| ViT-H-14 | 77.95 | 76.30 |

Table 1: **MLPs mean-ablation.** We simultaneously replace all the direct effects of the MLPs with their average taken across ImageNet's validation set. This results in only a small reduction in zero-shot classification performance.

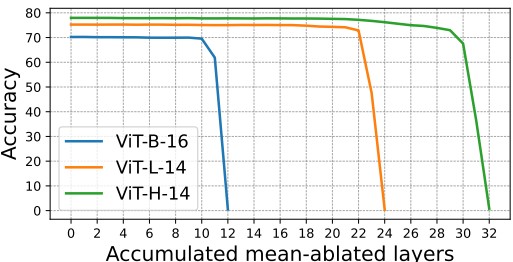

Figure 2: **MSAs accumulated mean-ablation.** We replace all the direct effects of the MSAs up to a given layer with their average taken across the ImageNet validation set. Only the replacement of the last few layers causes a large decrease in accuracy.

of the ViT output. By unrolling Eq. 3 across layers, the image representation can be written as:

$$M_{\text{image}}(I) = P\text{ViT}(I) = P\left[Z^0\right]_{cls} + \underbrace{\sum_{l=1}^{L} P\left[\text{MSA}^l(Z^{l-1})\right]_{cls}}_{\text{MSA terms}} + \underbrace{\sum_{l=1}^{L} P\left[\text{MLP}^l(\hat{Z}^l)\right]_{cls}}_{\text{MLP terms}} \quad (4)$$

Eq. 4 decomposes the image representation into *direct contributions* of MLPs, MSAs, and the input class token, allowing us to analyze each term separately. We ignore here the *indirect effects* of the output of one layer on another downstream layer. We use this decomposition (and further decompositions) to analyze CLIP's representations in the next sections.

**Evaluating the direct contribution of layers.** As a preliminary investigation, we study which of the components in Eq. 4 significantly affect the final image representation, and find that the large majority of the direct effects come from the *late attention layers*.

To study the direct effect of a component (or set of components), we use mean-ablation (Nanda et al., 2023), which replaces the component with its mean value across a dataset of images. Specifically, we measure the drop in zero-shot accuracy on a classification task before and after ablation. Components with larger direct effects should result in larger accuracy drops.

In our experiments, we compute means for each component over the ImageNet (IN) validation set and evaluate the drop in IN classification accuracy. We analyze the OpenCLIP ViT-H-14, L-14, and B-16 models (Ilharco et al., 2021), which were trained on LAION-2B (Schuhmann et al., 2022).

**MLPs have a negligible direct effect.** Table 1 presents the results of simultaneously mean-ablating all the MLPs. The MLPs do not have a significant direct effect on the image representation, as ablating all of them leads to only a small drop in accuracy (1%-3%).

**Only the last MSAs have a significant direct effect.** We next evaluate the direct effect of different MSA layers. To do so, we mean-ablate all MSA layers up to some layer $l$. Figure 2 presents the results: removing all the early MSA layers (up to the last 4) does not change the accuracy significantly. Mean-ablating these final MSAs, on the other hand, reduces the performance drastically.

In summary, the direct effect on the output is concentrated in the last 4 MSA layers. We therefore focus only on these layers in our subsequent analysis, ignoring the MLPs and the early MSA layers.

### 3.3 FINE-GRAINED DECOMPOSITION INTO HEADS AND POSITIONS

We present a more fine-grained decomposition of the MSA blocks that will be used in the next two sections. We focus on the output at the class token, as that is the only term appearing in Eq. 4. Following Elhage et al. (2021), we write the MSA output as a sum over $H$ independent attention heads and the $N$ input tokens:

$$\left[\text{MSA}^l(Z^{l-1})\right]_{cls} = \sum_{h=1}^{H} \sum_{i=0}^{N} x_i^{l,h}, \quad x_i^{l,h} = \alpha_i^{l,h} W_{VO}^{l,h} z_i^{l-1} \quad (5)$$

where $W_{VO}^{l,h} \in \mathbb{R}^{d \times d}$ are transition matrices and $\alpha_i^{l,h} \in \mathbb{R}$ are the attention weights from the class token to the $i$-th token ($\sum_{i=0}^{N} \alpha_i^{l,h} = 1$). Therefore, the MSA output can be decomposed into direct effects of individual heads and tokens.

| L21.H11 ("Geo-locations") | L23.H10 ("Counting") | L22.H8 ("Letters") |
|---|---|---|
| Photo captured in the Arizona desert | Image with six subjects | A photo with the letter V |
| Picture taken in Alberta, Canada | Image with four people | A photo with the letter F |
| Photo taken in Rio de Janeiro, Brazil | An image of the number 3 | A photo with the letter D |
| Picture taken in Cyprus | An image of the number 10 | A photo with the letter T |
| Photo taken in Seoul, South Korea | The number fifteen | A photo with the letter X |
| **L22.H11** ("Colors") | **L22.H6** ("Animals") | **L22.H3** ("Objects") |
| A charcoal gray color | Curious wildlife | An image of legs |
| Sepia-toned photograph | Majestic soaring birds | A jacket |
| Minimalist white backdrop | An image with dogs | A helmet |
| High-contrast black and white | Image with a dragonfly | A scarf |
| Image with a red color | An image with cats | A table |
| **L23.H12** ("Textures") | **L22.H1** ("Shapes") | **L22.H2** ("Locations") |
| Artwork with pointillism technique | A semicircular arch | Urban park greenery |
| Artwork with woven basket design | An isosceles triangle | Cozy home interior |
| Artwork featuring barcode arrangement | An oval | Urban subway station |
| Image with houndstooth patterns | Rectangular object | Energetic street scene |
| Image with quilted fabric patterns | A sphere | Tranquil boating on a lake |

Table 2: **Top-5 text descriptions extracted per head by our algorithm.** Top 5 components returned by TEXTSPAN applied to ViT-L, for several selected heads. See Section A.5 for results on all the heads.

Plugging the MSA output definition in Eq. 5 into the MSA term in Eq. 4, we obtain:

$$\sum_{l=1}^{L} P\left[\text{MSA}^l(Z^{l-1})\right]_{cls} = \sum_{l=1}^{L}\sum_{h=1}^{H}\sum_{i=0}^{N} c_{i,l,h}, \quad c_{i,l,h} = Px_i^{l,h} \tag{6}$$

In other words, the total direct effect of all attention blocks is the result of contracting the tensor $c$ across all of its dimensions. By contracting along only some dimensions, we can decompose effects in a variety of useful ways. For instance, we can contract along the spatial dimension $i$ to get a contribution for each head: $c_{\text{head}}^{l,h} = \sum_{i=0}^{N} c_{i,l,h}$. Alternatively, we can contract along layers and heads to get a contribution from each image token: $c_{\text{token}}^i = \sum_{l=1}^{L}\sum_{h=1}^{H} c_{i,l,h}$.

The quantities $c_{i,l,h}$, $c_{\text{head}}^{l,h}$ and $c_{\text{token}}^i$ all live in the $d'$-dimensional joint text-image representation space, which allows us to interpret them via text. For instance, given text description $t$, the quantity $\langle M_{\text{text}}(t), c_{\text{head}}^{l,h}\rangle$ intuitively measures the similarity of that head's output to description $t$.

## 4 DECOMPOSITION INTO ATTENTION HEADS

Motivated by the findings in Section 3.2, we turn to understanding the late MSA layers in CLIP. We use the decomposition into individual attention heads (Section 3.3), and present an algorithm for labeling the latent directions of each head with text descriptions. Examples of this labeling are depicted in Table 2 and Figure 4, with the labeling for all 64 late attention heads given in Section A.5.

Our labeling reveals that some heads exhibit specific semantic roles, e.g. "counting" or "location", in which many latent directions in the head track different aspects of that role. We show how to exploit these labeled roles both for property-specific image retrieval and for reducing spurious correlations.

### 4.1 TEXT-INTERPRETABLE DECOMPOSITION INTO HEADS

We decompose an MSA's output into text-related directions in the joint representation space. We rely on two key properties: First, the output of each MSA block is a sum of contributions of individual attention heads, as demonstrated in Section 3.3. Second, these contributions lie in the joint text-image representation space and so can be associated with text.

Recall from Section 3.3 that the MSA terms of the image representation (Eq. 4) can be written as a sum over heads, $\sum_{l,h} c_{\text{head}}^{l,h}$. To interpret a head's contribution $c_{\text{head}}^{l,h}$, we will find a set of text descriptions that explain most of the variation in the head's output (the head "principal components").

To formalize this, we take input images $I_1, ..., I_K$ with associated head outputs $c_1, ..., c_K$ (for simplicity, we fix the layer $l$ and head $h$ and omit it from the notation). As $c_1, ..., c_K$ are vectors in the

---

**Algorithm 1:** TEXTSPAN

---

**Input:** Head $(l, h)$ contribution $c_{\text{head}}^{l,h}$ for $K$ images stacked as rows in a matrix $C \in \mathbb{R}^{K \times d'}$, a pool of $M$
 text descriptions $\{t_i\}_{i=1}^M$, their corresponding CLIP text representations $R \in \mathbb{R}^{M \times d'}$ (projected to
 the head output space), and basis size $m$

**Output:** A set of text descriptions $\mathcal{T}$ and projected representations $C' \in \mathbb{R}^{K \times d'}$

**Initialization:** $C' \leftarrow \mathbf{0}_{K \times d'}, \mathcal{T} \leftarrow \phi$

**for** $i$ in $[1, ..., m]$ **do**
$\quad D \leftarrow RC^T$
$\quad j^* \leftarrow \arg\max_{j=1}^M \text{Var}(D[j])$
$\quad \mathcal{T} \leftarrow \mathcal{T} \cup \{t_{j^*}\}$
$\quad$**for** $k$ in $[1, ..., K]$ **do**
$\quad\quad C'[k] \leftarrow C'[k] + \frac{\langle C[k], R[j^*]\rangle}{||R[j^*]||^2} R[j^*]$
$\quad\quad C[k] \leftarrow C[k] - \frac{\langle C[k], R[j^*]\rangle}{||R[j^*]||^2} R[j^*]$
$\quad$**for** $k$ in $[1, ..., M]$ **do**
$\quad\quad R[k] \leftarrow R[k] - \frac{\langle R[k], R[j^*]\rangle}{||R[j^*]||^2} R[j^*]$

---

joint text-image space, each text input $t$ defines a direction $M_{\text{text}}(t)$ in that space. Given a collection of text directions $\mathcal{T}$, let $\text{Proj}_{\mathcal{T}}$ denote the projection onto the span of $\{M_{\text{text}}(t) \mid t \in \mathcal{T}\}$. We define the *variance explained by* $\mathcal{T}$ as the variance under this projection:

$$V_{\text{explained}}(\mathcal{T}) = \frac{1}{K} \sum_{k=1}^K \| \text{Proj}_{\mathcal{T}}(c_k - c_{\text{avg}})\|_2^2, \text{ where } c_{\text{avg}} = \frac{1}{K} \sum_{k=1}^K c_k. \tag{7}$$

We aim to find a set of $m$ descriptions $\mathcal{T}$ for each head that maximizes $V_{\text{explained}}(\mathcal{T})$. Unlike regular PCA, there is no closed-form solution to this optimization problem, so we take a greedy approach.

**Greedy algorithm for descriptive set mining.** To approximately maximize the explained variance in Eq. 7, we start with a large pool of candidate descriptions $\{t_i\}_{i=1}^M$ and greedily select from it to obtain the set $\mathcal{T}$.

Our algorithm, TEXTSPAN, is presented in Alg. 1. It starts by forming the matrix $C \in \mathbb{R}^{K \times d'}$ of outputs for head $(l, h)$, as well as the matrix $R \in \mathbb{R}^{M \times d'}$ of representations for the candidate descriptions, projected onto the span of $C$. In each round, TEXTSPAN computes the dot product between each row of $R$ and the head outputs $C$, and finds the row with the highest variance $R[j^*]$ (the first "principle component"). It then projects that component away from all rows and repeats the process to find the next components. The projection step ensures that each new component adds variance that is orthogonal to the earlier components.

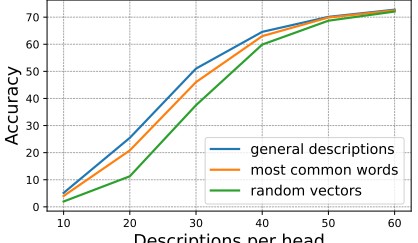

Figure 3: ImageNet classification accuracy for the image representation projected to TEXTSPAN bases. We evaluate our algorithm for different initial description pools, and with different output sizes.

TEXTSPAN requires an initial set of descriptions $\{t_i\}_{i=1}^M$ that is diverse enough to capture the output space of each head. We use a set of sentences that were generated by prompting ChatGPT-3.5 to produce general image descriptions. After obtaining an initial set, we manually prompt ChatGPT to generate more examples of specific patterns we found (e.g. texts that describe more colors). This results in 3498 sentences. In our experiments, we also consider two simpler baselines—one-word descriptions comprising the most common words in English, and a set of random $d'$-dimensional vectors that do not correspond to text (see Section A.3 for the ChatGPT prompt and more details about the baselines).

## 4.2 EXPERIMENTS

We apply TEXTSPAN to find a basis of text descriptions for all heads in the last 4 MSA layers. We first verify that this set captures most of the model's behavior and that text descriptions track

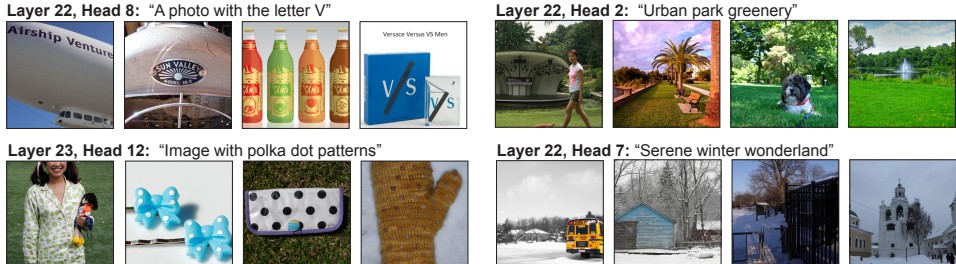

Figure 4: **Top-4 images for the top head description found by TEXTSPAN.** We retrieve images with the highest similarity score between $c_{\text{head}}^{l,h}$ and the top text representation found by TEXTSPAN. They correspond to the provided text descriptions. See Figure 11 in the appendix for randomly selected heads.

image properties. We then show that some heads are responsible for capturing specific image properties (see Figure 1(1)). We use this finding for two applications—reducing known spurious cues in downstream classification and property-specific image retrieval.

**Experimental setting.** We apply TEXTSPAN to all the heads in the last 4 layers of CLIP ViT-L, which are responsible for most of the direct effects on the image representation (see Section 3.2). We consider a variety of output sizes $m \in \{10, 20, 30, 40, 50, 60\}$.

We first verify that the resulting text representations capture the important variation in the image representation, as measured by zero-shot accuracy on ImageNet. We simultaneously replace each head's direct contribution $c_{\text{head}}^{l,h}$ with its projection to the text representations $\text{Proj}_{\mathcal{T}(l,h)} c_{\text{head}}^{l,h}$ (where $\mathcal{T}(l,h)$ is the obtained text set for head $(l,h)$). We also mean-ablate all other terms in the representation (MLPs and the rest of the MSA layers).

The results are shown in Fig. 3: 60 descriptions per head suffice to reach 72.77% accuracy (compared to 75.25% base accuracy). Moreover, using our ChatGPT-generated descriptions as the candidate pool yields higher zero-shot accuracy than either common words or random directions, for all the different sizes $m$. In summary, we can approximate CLIP's representation by projecting each head output, a 768-dimensional vector, to a (head-specific) 60-dimensional text-interpretable subspace.

**Some attention heads capture specific image properties.** We report selected head descriptions from TEXTSPAN ($m = 60$) in Table 2, with full results in Appendix A.5. For some heads, the top descriptions center around a single image property like texture (L23H12), shape (L22H1), object count (L23H10), and color (L22H11). This suggests that these heads capture *specific image properties*. We qualitatively verify that the text tracks these image properties by retrieving the images with the largest similarity $\langle M_{\text{text}}(t_i), c_{\text{head}}^{l,h} \rangle$ for the top extracted text descriptions $t_i$. The results in Fig. 4 and 11 show that the returned images indeed match the text.

**Reducing known spurious cues.** We can use our knowledge of head-specific roles to manually remove spurious correlations. For instance, if location is being used as a spurious feature, we can ablate heads that specialize in geolocation to hopefully reduce reliance on the incorrect feature.

We validate this idea on the Waterbirds dataset (Sagawa et al., 2019), which combines waterbird and landbird photographs from the CUB dataset Welinder et al. (2010) with image backgrounds (water/land background) from the Places dataset (Zhou et al., 2016). Here image background is a spurious cue, and models tend to misclassify waterbirds on land backgrounds (and vice versa).

To reduce spurious cues, we manually annotated the role of each head using the text descriptions from TEXTSPAN, mean-ablated the direct contributions of all "geolocation" and "image-location" heads, and then evaluated the zero-shot accuracy on Waterbirds, computing the worst accuracy across subgroups as in Sagawa et al. (2019). As a baseline, we also ablated 10 random heads and reported the top accuracy out of 5 trials. As shown in Table 3, the worst-group accuracy increases by a large margin—by 25.2% for ViT-L. This exemplifies that the head roles we found with TEXTSPAN help us to design representations with less spurious cues, without any additional training.

**Property-based image retrieval.** Since some heads specialize to image properties, we can use their representations to obtain a property-specific similarity metric. To illustrate this, for a given head $(h, l)$, we compute the inner product $\langle c_{\text{head}}^{l,h}(I), c_{\text{head}}^{l,h}(I') \rangle$ between a base image $I$ and all other

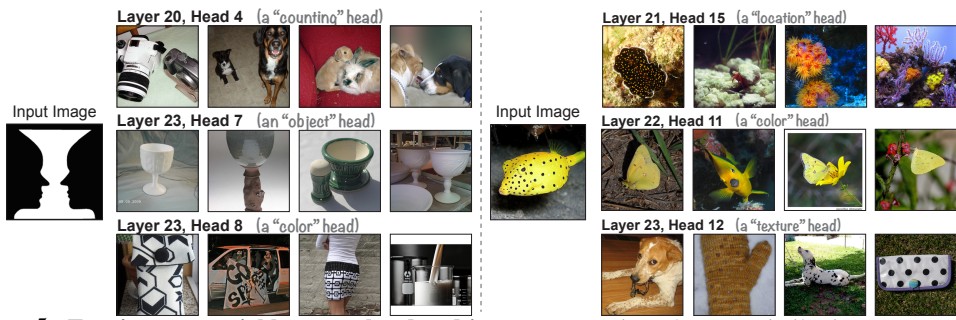

Figure 5: **Top-4 nearest neighbors per head and image.** We retrieve the most similar images to an input image by computing the similarity of the direct contributions of individual heads. As some heads capture specific aspects of the image (e.g. colors/objects), retrieval according to this metric results in images that are most similar regarding these aspects. See additional results in the project page and appendix (Fig. 12).

images in the dataset, retrieving the images with the highest similarity. Figure 5 shows the resulting nearest neighbors for heads that capture different properties. The retrieved images are different for each head and match the head-specific properties. In the left example, if we use a head that captures color for retrieval, the nearest neighbors are images with black-and-white objects. If we use a head that counts objects, the nearest neighbors are images with two objects.

## 5    DECOMPOSITION INTO IMAGE TOKENS

Decomposing the image representation across heads enabled us to answer *what* each head contributes to the output representation. We can alternately decompose the representation across image tokens to tell us *which image regions* contribute to the output for a given text direction $M_{\text{text}}(t)$. We find that these regions match the image parts that $t$ describes, thereby yielding a zero-shot semantic image segmenter. We compare this segmenter to existing CLIP-based zero-shot methods and find that it is state-of-the-art. Finally, we decompose each head's direct contributions into per-head image tokens and use this to obtain fine-grained visualizations of the information flow from input images to output semantic representations.

**Decomposing MSA outputs into image tokens.** Applying the decomposition from Section 3.3, if we group the terms $c_{i,l,h}$ by position $i$ instead of head $(l, h)$, we obtain the identity $M_{\text{image}}(I) = \sum_{i=0}^{N} c_{\text{token}}^i(I)$, where $c_{\text{token}}^i(I)$ is the sum of the output at location $i$ across all heads $(l, h)$. We empirically find that the contribution of the class token $c_{\text{token}}^0$ has negligible direct effect on zero-shot accuracy (see mean-ablation in A.2). Therefore, we focus on the $N$ image tokens.

We use the decomposition into image tokens to generate a heatmap that measures how much the output from each image position contributes to writing in a given text direction. Given a text description $t$, we obtain this heatmap by computing the score $\langle c_{\text{token}}^i(I), M_{\text{text}}(t)\rangle$ for each position $i$.

**Quantitative segmentation results.** We follow a standard protocol for evaluating heatmap-based explainability methods (Chefer et al., 2021). We first compute image heatmaps given descriptions of the image class (e.g. "An image of a {class}")[3]. We then binarize them (by applying a threshold) to obtain a foreground/background segmentation. We compare the segmentation quality to zero-shot segmentations produced by other explainability methods in the same manner.

We evaluate the methods on ImageNet-segmentation (Guillaumin et al., 2014), which contains a subset of 4,276 images from the ImageNet validation set with annotated segmentations. Table 4 displays the results: our decomposition is more accurate than existing methods across all metrics. See Chefer et al. (2021) for details about the compared methods and metrics, and additional qualitative comparisons in Section A.6.

**Joint decomposition into per-head image tokens.** Finally, we can jointly decompose the output of CLIP across both heads and locations. We use this decomposition to visualize what regions affect each of the basis directions found by TEXTSPAN. Recall that $c_{i,l,h}$ from Eq. 6 is the direct contribution of token $i$ at head $(h, l)$ to the representation. For each image token $i$, we take the inner

---

[3]We normalize out bias terms by subtracting from the heatmap an averaged heatmap computed across all class descriptions in ImageNet.

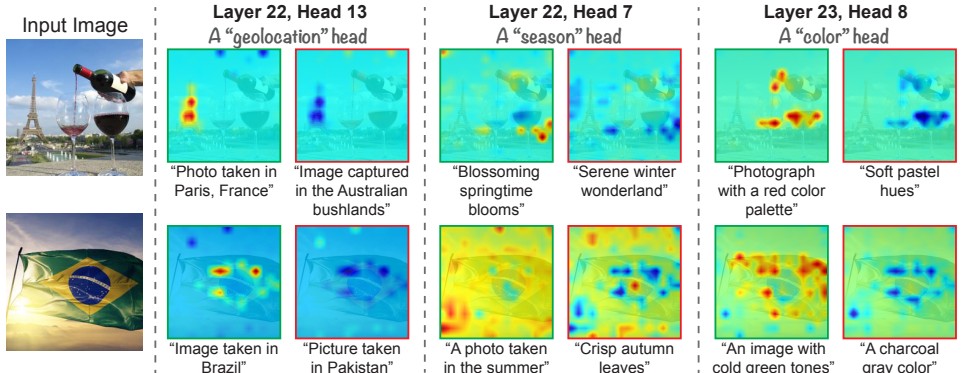

Figure 6: **Joint decomposition examples.** For each head $(l, h)$, the left heatmap (green border) corresponds to the description that is most similar to $c_{\text{head}}^{l,h}$ among the TEXTSPAN output set. The right heatmap (red border) corresponds to the least similar text in this set (for $m = 60$). See Figure 9 for more results.

products between $c_{i,l,h}$ and a basis direction $M_{\text{text}}(t)$ and obtain a *per-head* similarity heatmap. This visualizes the flow of information from input images to the text-labeled basis directions.

In Figure 6, we compute heatmaps for the two TEXTSPAN basis elements that have the largest and smallest (most negative) coefficients when producing each head's output. The highlighted regions match the text description for that basis direction—for instance, L22H13 is a geolocation head, its highest-activating direction for the top image is "Photo taken in Paris, France", and the image tokens that contribute to this direction are those matching the Eiffel Tower.

|  | base | top random | ours |
|---|---|---|---|
| ViT-B-16 | 45.6 | 52.3 | **57.5** |
| ViT-L-14 | 47.7 | 57.7 | **72.9** |
| ViT-H-14 | 37.2 | 37.0 | **43.3** |

Table 3: **Worst-group accuracy on Waterbirds.** We reduce spurious cues by ablating property-specific heads. See Tables 11-14 for fine-grained results.

|  | Pixel Acc. ↑ | mIoU ↑ | mAP ↑ |
|---|---|---|---|
| LRP (Binder et al., 2016) | 52.81 | 33.57 | 54.37 |
| partial-LRP (Voita et al., 2019) | 61.49 | 40.71 | 72.29 |
| rollout (Abnar & Zuidema, 2020) | 60.63 | 40.64 | 74.47 |
| raw attention | 65.67 | 43.83 | 76.05 |
| GradCAM Selvaraju et al. (2017) | 70.27 | 44.50 | 70.30 |
| Chefer et al. (2021) | 69.21 | 47.47 | 78.29 |
| Ours | **75.21** | **54.50** | **81.61** |

Table 4: **Segmentation performance on ImageNet-segmentation.** The image tokens decomposition results in significantly more accurate zero-shot segmentation than previous methods.

## 6 LIMITATIONS AND DISCUSSION

We studied CLIP's image representation by analyzing how individual model components affect it. Our findings allowed us to reduce spurious cues in downstream classification and improve zero-shot segmentation. We present two limitations of our investigation and conclude with future directions.

**Indirect effects.** We analyzed only the direct effects of model components on the representation. Studying indirect effects (e.g. information flow from early layers to deeper ones) can provide additional insights into the internal structure of CLIP and unlock more downstream applications.

**Not all attention heads have clear roles.** The outputs of TEXTSPAN show that not every head captures a single image property (see results in Section A.5). We consider three possible explanations for this: First, some heads may not correspond to coherent properties. Second, the initial descriptions pool does not include descriptions of any image property. Third, some heads may collaborate and have a coherent role only when their outputs are addressed together. Uncovering the roles of more complex structures in CLIP can improve the performance of the described applications.

**Future work.** We believe that similar analysis for other CLIP architectures (e.g. ResNet) can shed light on the differences between the output representations of different networks. Moreover, our insights may help to design better CLIP image encoder architectures and feature extractors for downstream tasks. We plan to explore these directions in future work.

**Acknowledgements.** We would like to thank Jean-Stanislas Denain for the insightful discussions and comments. We thank Jixahai Feng, Fred Zhang, and Erik Jones for helpful feedback on the manuscript. YG is supported by the Google Fellowship. AE is supported in part by DoD, including DARPA's MCS and ONR MURI, as well as funding from SAP. JS is supported by the NSF Awards No. 1804794 & 2031899.

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

# A    APPENDIX

## A.1    LAYER NORMALIZATION

We describe here the modifications that are needed to be incorporated in our method to take into account layer-normalizations. There are two places where layer-normalizations are used - before the projection layer (to the output of the ViT), and before each layer in the ViT (to the layer input). We present how the individual contributions of $c_{i,l,h}$, $c_{\text{head}}^{l,h}$ and $c_{\text{token}}^i$ should be changed.

**Pre-projection layer normalization.** As mentioned in the Section 3.1, in many implementations of CLIP, a layer-normalization $LN$ is applied to the output of the ViT before the projection layer. Formally, the image representation of image $I$ is then:

$$M_{\text{image}}(I) = P\mathsf{LN}(\mathsf{ViT}(I)) \tag{8}$$

The normalization layer can be rewritten as:

$$\mathsf{LN}(x) = \gamma * \frac{x - \mu_l}{\sqrt{\sigma_l^2 + \epsilon}} + \beta = \left[ \frac{\gamma}{\sqrt{\sigma_l^2 + \epsilon}} \right] * x - \left[ \frac{\mu_l \gamma}{\sqrt{\sigma_l^2 + \epsilon}} - \beta \right] \tag{9}$$

where $x \in \mathbb{R}^d$ is the input token, $\mu_l, \sigma_l \in \mathbb{R}$ are the mean and standard deviation, and $\gamma, \beta \in \mathbb{R}^d$ are learned vectors. To incorporate the layer normalization in our decomposition, we compute the mean and the standard deviation during the forward pass of the model. The multiplicative term, $\frac{\gamma}{\sqrt{\sigma_l^2 + \epsilon}}$ is absorbed into the projection matrix $P$. The contribution of $\frac{\mu_l \gamma}{\sqrt{\sigma_l^2 + \epsilon}} - \beta$ is split equally between all the $c_{i,l,h}$ terms in the Eq. 6. We apply these modifications when we decompose OpenCLIP-based models.

**MLPs and MSAs input layer normalizations.** In the main paper, we do not describe the normalization layers that are applied to each input of MLP and MSA in the model. More accurately, the residual updates of the ViT are:

$$\hat{Z}^l = \mathsf{MSA}^l(\mathsf{LN}^l(Z^{l-1})) + Z^{l-1}, \quad Z^l = \mathsf{MLP}^l(\hat{\mathsf{LN}}^l(\hat{Z}^l)) + \hat{Z}^l \tag{10}$$

Where $\hat{\mathsf{LN}}^l$ and $\mathsf{LN}^l$ are the layer normalizations applied to each token in the input matrix of the MLP layers and MSA layers. This modification does not affect our corollaries about the direct contributions of the MLP layers and MSA layers, as we only address the outputs of these layers. The only other equation in which this modification takes place is in Eq. 5:

$$\left[ \mathsf{MSA}^l(Z^{l-1}) \right]_{cls} = \sum_{h=1}^{H} \sum_{i=0}^{N} x_i^{l,h}, \quad x_i^{l,h} = \alpha_i^{l,h} \mathsf{LN}^l(z_i^{l-1}) W_{VO}^{l,h} \tag{11}$$

## A.2    MEAN-ABLATION OF THE CLASS-TOKEN ATTENDED FROM ITSELF

We show that we can ignore the direct effect of the class token in the MSAs term when we decompose it into tokens (see section 5). We mean-ablate the direct contribution of the class token to the MSAs term in Eq. 6. We simultaneously ablate both the class token and the MLPs. The ImageNet zero-shot classification performances of the three ViT models are shown in Table 5. As shown, the direct contributions of all the MLP layers *and* the direct contributions of the class token in the decomposed MSAs term results in a negligible drop in performance for all the models.

## A.3    TEXT DESCRIPTIONS

**General text descriptions.** To generate the set of text descriptions that are used by our algorithm, we prompted ChatGPT (GPT-3.5) to produce image descriptions. We used the prompt provided in Table 6, and manually prompted the language model to generate more examples for specific patterns we found in the initial result (e.g. more colors, more letters). This process resulted in 3498 sentences.

**Most common words.** For the set of most common words, we used the same number of examples, and took the 3498 most common English words, as determined by n-gram frequency analysis of Google's Trillion Word Corpus (Segaran & Hammerbacher (2009)).

|          | Base accuracy | + class token ablation | + MLPs ablation |
|----------|---------------|------------------------|-----------------|
| ViT-B-16 | 70.22         | 69.37                  | 67.32           |
| ViT-L-14 | 75.25         | 74.38                  | 73.87           |
| ViT-H-14 | 77.95         | 76.89                  | 76.29           |

Table 5: **Mean-ablation of the class token contribution to the MSAs term.** The overall drop in accuracy is relatively small, even when the MLPs are replaced by their mean across ImageNet validation set.

**Class-specific text descriptions.** We generate additional class-specific text descriptions, by prompting ChatGPT with the prompt template provided in table 6. We queried to model for each of the ImageNet class names. This process resulted in 28767 unique sentences.

**Random vectors.** As a baseline we created a random set of 3498 vectors sampled from a unit Gaussian.

| **General text descriptions initial prompt** |
|---|
| Imagine you are trying to explain a photograph by providing a complete set of image characteristics. Provide generic image characteristics. Be as general as possible and give short descriptions presenting one characteristic at a time that can describe almost all the possible images of a wide range of categories. Try to cover as many categories as possible, and don't repeat yourself. Here are some possible phrases: "An image capturing an interaction between subjects", "Wildlife in their natural habitat", "A photo with a texture of mammals", "An image with cold green tones", "Warm indoor scene", "A photo that presents anger". Just give the short titles, don't explain why, and don't combine two different concepts (with "or" or "and"). Make each item in the list short but descriptive. Don't be too specific. |
| **Class-specific text descriptions prompt** |
| Provide 40 image characteristics that are true for almost all the images of {*class*}. Be as general as possible and give short descriptions presenting one characteristic at a time that can describe almost all the possible images of this category. Don't mention the category name itself (which is "{*class*}"). Here are some possible phrases: "Image with texture of ...", "Picture taken in the geographical location of...", "Photo that is taken outdoors", "Caricature with text", "Image with the artistic style of...", "Image with one/two/three objects", "Illustration with the color palette ...", "Photo taken from above/below", "Photograph taken during ... season". Just give the short titles, don't explain why, and don't combine two different concepts (with "or" or "and"). |

Table 6: **ChatGPT prompts for image descriptions generation.**

## A.4 ADDITIONAL INITIAL DESCRIPTION POOL ABLATION

We present additional ablation of the initial set of text descriptions provided to TEXTSPAN. The text description generation processes for each of the pools are described in Section A.3.

As shown in Figure 7, using the class-specific descriptions pool that includes around $\times 8$ more examples than the general descriptions pool, allows us to obtain higher accuracy with fewer descriptions per head (smaller $m$). Nevertheless, using each of the two pools results in relatively similar accuracy with $m = 60$.

## A.5 TEXTSPAN OUTPUTS FOR CLIP-VIT-L

We apply TEXTSPAN to the attention heads of the last 4 layers of CLIP ViT-L. Tables 7-10 present the first 5 descriptions per head.

## A.6 QUALITATIVE RESULTS FOR IMAGE TOKEN DECOMPOSITION

Figure 8(a) shows the similarity heatmaps for text descriptions. As presented our heatmaps highlight the objects that are described in the text. Figure 8(b) presents the relative similarity heatmaps

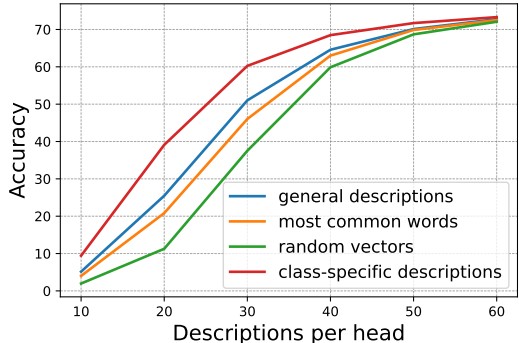

Figure 7: ImageNet classification accuracy for the image representation projected to TEXTSPAN bases (additional results). We evaluate our algorithm for different initial description pools, and with different output sizes.

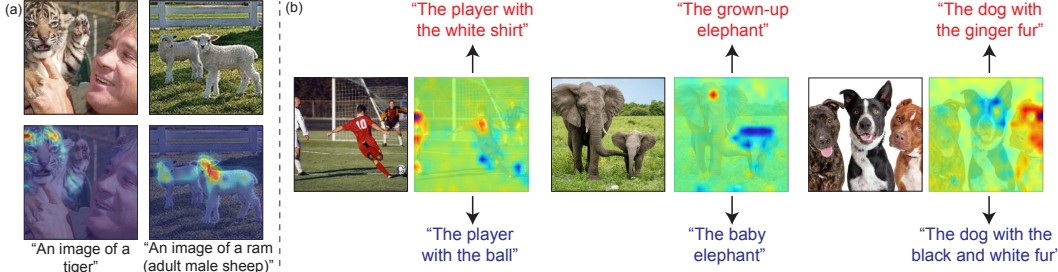

Figure 8: **Heatmaps produced by the image token decomposition**. We visualize (a) what areas in the image directly contribute to the similarity score between the image representation and a text representation and (b) what areas make an image representation more similar to one text representation rather than another.

given two descriptions (by subtracting between the two heatmaps). The areas in the images that make the image representations more similar to one of the text representations rather than the other, correspond to the areas that are mentioned by it and ignored by the other text.

## A.7    MOST SIMILAR IMAGES TO TEXTSPAN RESULTS

We randomly choose 3 attention heads from the last 4 layers of CLIP ViT-L. For each head $(l, h)$, we retrieve the 3 images with the highest similarity score between their $c_{\text{head}}^{l,h}$ and the top 10 text representations found by our algorithm. The retrieval is done from ImageNet validation set. The results are presented in Figure 11. As shown, in most cases, the top text representation corresponds to the attributes of the images.

| Layer 20, Head 0 | Layer 20, Head 1 |
|---|---|
| Picture taken in Hungary | Picture taken in Seychelles |
| Image taken in New England | Picture taken in Saudi Arabia |
| Futuristic technological concept | Muted urban tones |
| Playful siblings | Man-made pattern |
| Picture taken in the English countryside | an image of glasgow |
| **Layer 20, Head 2** | **Layer 20, Head 3** |
| Image of a police car | Intrica wood carvingte |
| Picture taken in Laos | Image snapped in Spain |
| Remote alpine chalet | Photo taken in Bora Bora, French Polynesia |
| A photograph of a small object | An image of a Preschool Teacher |
| Desert sandstorm | A breeze |
| **Layer 20, Head 4** | **Layer 20, Head 5** |
| Image with a pair of subjects | an image of samoa |
| Image with five subjects | Urban nostalgia |
| Image with a trio of friends | A photo with the letter K |
| A photo of an adult | Image snapped in the Colorado Rockies |
| Image with a seven people | Serendipitous discovery |
| **Layer 20, Head 6** | **Layer 20, Head 7** |
| Bustling city square | Energetic children |
| Peaceful village alleyway | Grumpy facial expression |
| ornate cathedral | Intricate ceramic patterns |
| Image taken in the Alaskan wilderness | Photo taken in Bangkok, Thailand |
| Modern airport terminal | Subdued moments |
| **Layer 20, Head 8** | **Layer 20, Head 9** |
| Photo taken in Rioja, Spain | Tranquil Asian temple |
| Photo taken in Borneo | Vibrant city nightlife |
| Vibrant urban energy | A photo with the letter R |
| Picture captured in the Icelandic glaciers | intricate mosaic artwork |
| serene oceanside scene | Photo taken in the Rub' al Khali (Empty Quarter) |
| **Layer 20, Head 10** | **Layer 20, Head 11** |
| A bowl | Photo taken in Beijing, China |
| A bottle | Photo with retro color filters |
| Nostalgic pathways | Image with holographic cyberpunk aesthetics |
| A laptop | Urban street fashion |
| Reflective ocean view | Photograph with the artistic style of tilt-shift |
| **Layer 20, Head 12** | **Layer 20, Head 13** |
| Photo with grainy, old film effect | Image taken from a distance |
| Detailed illustration | Photograph with the artistic style of split toning |
| Serene beach sunset | Photo taken in Beijing, China |
| An image of the number 10 | A close-up shot |
| An image of the number 5 | An image of a Novelist |
| **Layer 20, Head 14** | **Layer 20, Head 15** |
| Quirky street performer | Remote hilltop hut |
| Antique sculptural element | Photo taken in Barcelona, Spain |
| Celebratory atmosphere | Dynamic movement |
| Overwhelmed facial expression | Caricature of an influential leader |
| Serene winter wonderland | A picture of Samoa |

Table 7: **Top-5 results of TEXTSPAN.** Applied to the heads at layer 20 of CLIP-ViT-L.

| Layer 21, Head 0 | Layer 21, Head 1 |
|---|---|
| Timeless black and white | Picture taken in the southeastern United States |
| Vintage sepia tones | Picture taken in the Netherlands |
| Image with a red color | Image taken in Brazil |
| A charcoal gray color | Image captured in the Australian bushlands |
| Soft pastel hues | Picture taken in the English countryside |
| **Layer 21, Head 2** | **Layer 21, Head 3** |
| A photo of a woman | Precise timekeeping mechanism |
| A photo of a man | Image snapped in the Canadian lakes |
| Energetic children | An image of Andorra |
| An image with dogs | thrilling sports challenge |
| A picture of a baby | Photo taken in Namib Desert |
| **Layer 21, Head 4** | **Layer 21, Head 5** |
| An image with dogs | Inquisitive facial expression |
| A picture of a bridge | Artwork featuring typographic patterns |
| A photo with the letter R | A photograph of a big object |
| Dramatic skies | Reflective landscape |
| Ancient castle walls | Burst of motion |
| **Layer 21, Head 6** | **Layer 21, Head 7** |
| Photo taken in the Italian pizzerias | A pin |
| thrilling motorsport race | A thimble |
| Urban street fashion | A bookmark |
| An image of a Animal Trainer | Picture taken in Rwanda |
| Serene countryside sunrise | A pen |
| **Layer 21, Head 8** | **Layer 21, Head 9** |
| Inviting coffee shop | Photograph with a blue color palette |
| Photograph taken in a music store | Image with a purple color |
| An image of a News Anchor | Image with a pink color |
| Joyful family picnic scene | Image with a orange color |
| cozy home library | Timeless black and white |
| **Layer 21, Head 10** | **Layer 21, Head 11** |
| Playful winking facial expression | Photo captured in the Arizona desert |
| Joyful toddlers | Picture taken in Alberta, Canada |
| Close-up of a textured plastic | Photo taken in Rio de Janeiro, Brazil |
| An image of a Teacher | Picture taken in Cyprus |
| Image with a seven people | Photo taken in Seoul, South Korea |
| **Layer 21, Head 12** | **Layer 21, Head 13** |
| Photo with grainy, old film effect | Quiet rural farmhouse |
| Macro botanical photography | Lively coastal fishing port |
| A laptop | an image of liechtenstein |
| Vintage nostalgia | Image taken in the Florida Everglades |
| serene mountain retreat | thrilling motorsport race |
| **Layer 21, Head 14** | **Layer 21, Head 15** |
| Photo taken in Beijing, China | Submerged underwater scene |
| Cheerful adolescents | Artwork featuring overlapping scribbles |
| Picture taken in Ecuador | Surrealist artwork with dreamlike elements |
| Dreamy haze | Serene winter wonderland |
| Image captured in the Greek islands | Wildlife in their natural habitat |

Table 8: **Top-5 results of TEXTSPAN.** Applied to the heads at layer 21 of CLIP-ViT-L.

| Layer 22, Head 0 | Layer 22, Head 1 |
|---|---|
| Artwork with pointillism technique | A semicircular arch |
| Artwork with woven basket design | An isosceles triangle |
| Artwork featuring barcode arrangement | An oval |
| Image with houndstooth patterns | Rectangular object |
| Image with quilted fabric patterns | A sphere |
| **Layer 22, Head 2** | **Layer 22, Head 3** |
| Urban park greenery | An image of legs |
| cozy home interior | A jacket |
| Urban subway station | A helmet |
| Energetic street scene | A scarf |
| Tranquil boating on a lake | A table |
| **Layer 22, Head 4** | **Layer 22, Head 5** |
| An image with dogs | Harmonious color scheme |
| Joyful toddlers | An image of cheeks |
| Serene waterfront scene | Vibrant vitality |
| thrilling sports action | Captivating scenes |
| A picture of a baby | Dramatic chiaroscuro photography |
| **Layer 22, Head 6** | **Layer 22, Head 7** |
| Curious wildlife | Serene winter wonderland |
| Majestic soaring birds | Blossoming springtime blooms |
| An image with dogs | Crisp autumn leaves |
| Image with a dragonfly | A photo taken in the summer |
| An image with cats | Posed shot |
| **Layer 22, Head 8** | **Layer 22, Head 9** |
| A photo with the letter V | A photo of food |
| A photo with the letter F | delicate soap bubble play |
| A photo with the letter D | Dynamic and high-energy music performance |
| A photo with the letter T | Hands in an embrace |
| A photo with the letter X | Futuristic technology display |
| **Layer 22, Head 10** | **Layer 22, Head 11** |
| Image with a yellow color | A charcoal gray color |
| Image with a orange color | Sepia-toned photograph |
| An image with cold green tones | Minimalist white backdrop |
| Image with a pink color | High-contrast black and white |
| Sepia-toned photograph | Image with a red color |
| **Layer 22, Head 12** | **Layer 22, Head 13** |
| Photo taken in Namib Desert | Image taken in Thailand |
| Ocean sunset silhouette | Picture taken in the Netherlands |
| Photo taken in the Brazilian rainforest | Picture taken in the southeastern United States |
| Serene countryside sunrise | Image captured in the Australian bushlands |
| Bustling cityscape at night | Picture taken in the geographical location of Spain |
| **Layer 22, Head 14** | **Layer 22, Head 15** |
| A silver color | contemplative urban view |
| Play of light and shadow | Photograph revealing frustration |
| Image with a white color | Celebratory atmosphere |
| A charcoal gray color | Captivating authenticity |
| Cloudy sky | Intense athletic competition |

Table 9: **Top-5 results of TEXTSPAN.** Applied to the heads at layer 22 of CLIP-ViT-L.

| Layer 23, Head 0 | Layer 23, Head 1 |
|---|---|
| Intrica wood carvingte | Photograph taken in a retro diner |
| Nighttime illumination | Intense athlete |
| Image with woven fabric design | Detailed illustration of a futuristic bioreactor |
| Image with shattered glass reflections | Image with holographic retro gaming aesthetics |
| A photo of food | Antique historical artifact |
| **Layer 23, Head 2** | **Layer 23, Head 3** |
| Image showing prairie grouse | Bustling city square |
| Image with a penguin | Serene park setting |
| A magnolia | Warm and cozy indoor scene |
| An image with dogs | Modern airport terminal |
| An image with cats | Remote hilltop hut |
| **Layer 23, Head 4** | **Layer 23, Head 5** |
| Playful siblings | Intertwined tree branches |
| A photo of a young person | Flowing water bodies |
| Image with three people | A meadow |
| A photo of a woman | A smoky plume |
| A photo of a man | Blossoming springtime blooms |
| **Layer 23, Head 6** | **Layer 23, Head 7** |
| Picture taken in Sumatra | A paddle |
| Picture taken in Alberta, Canada | A ladder |
| Picture taken in the geographical location of Spain | Intriguing and enigmatic passageway |
| Image taken in New England | A bowl |
| Photo captured in the Arizona desert | A table |
| **Layer 23, Head 8** | **Layer 23, Head 9** |
| Photograph with a red color palette | ornate cathedral |
| An image with cold green tones | detailed reptile close-up |
| Timeless black and white | Image with a seagull |
| Image with a yellow color | A clover |
| Photograph with a blue color palette | Futuristic space exploration |
| **Layer 23, Head 10** | **Layer 23, Head 11** |
| Image with six subjects | A photo with the letter N |
| Image with a four people | A photo with the letter J |
| An image of the number 3 | Serendipitous discovery |
| An image of the number 10 | A fin |
| The number fifteen | Unusual angle |
| **Layer 23, Head 12** | **Layer 23, Head 13** |
| Image with polka dot patterns | Photo taken in a museum |
| Striped design | Surreal digital collage |
| Checkered design | Cinematic portrait with dramatic lighting |
| Artwork with pointillism technique | Collage of vintage magazine clippings |
| Photo taken in Galápagos Islands | Candid documentary photography |
| **Layer 23, Head 14** | **Layer 23, Head 15** |
| An image with dogs | Resonant harmony |
| Majestic soaring birds | Subtle nuance |
| Graceful swimming fish | An image of cheeks |
| An image with bikes | emotional candid gaze |
| Picture with boats | Whimsicachildren's scenel |

Table 10: **Top-5 results of TEXTSPAN.** Applied to the heads at layer 23 of CLIP-ViT-L.

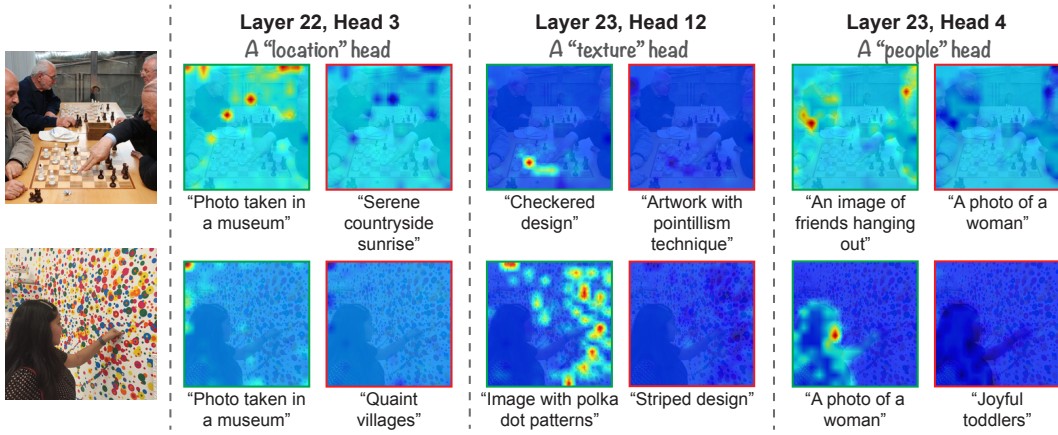

Figure 9: **Additional joint decomposition examples.**

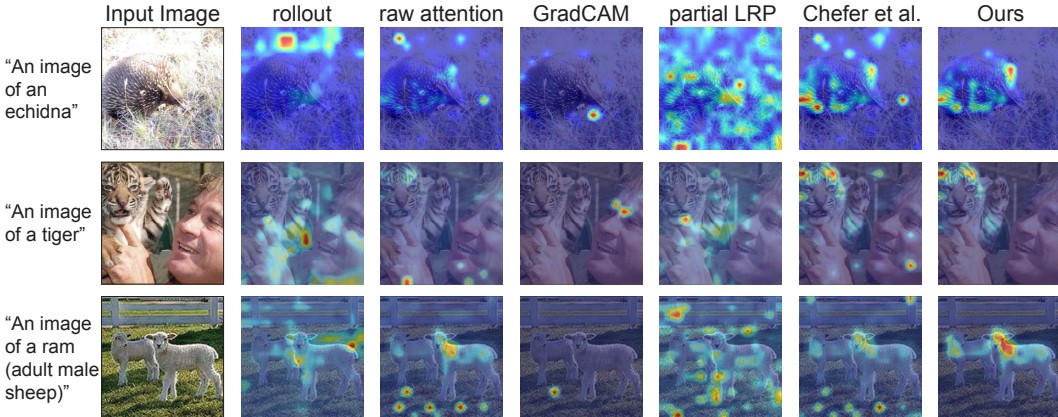

Figure 10: **Comparison to other explainability methods.** The highlighted regions produced by our decomposition are more aligned with the areas of the image that are mentioned in the text.

|          | base | ours     |
|----------|------|----------|
| ViT-B-16 | 76.7 | **83.8** |
| ViT-L-14 | 73.1 | **84.2** |
| ViT-H-14 | 77.0 | **84.1** |

Table 11: **Overall classification accuracy on Waterbirds dataset.** We reduce spurious cues by zeroing the direct effects of property-specific heads.

|                 | water background | land background  |
|-----------------|------------------|------------------|
| waterbird class | 92.1 (**93.1**)  | **77.8** (66.2)  |
| landbird class  | **72.9** (47.7)  | **94.9** (94.8)  |

Table 12: **Zero-shot classification accuracy on Waterbirds dataset, per class and background (ViT-L).** The accuracy for the baseline CLIP model is in parentheses. As shown, we reduce the spurious correlation between the background and the object class.

|                 | water background | land background  |
|-----------------|------------------|------------------|
| waterbird class | 62.3 (**69.8**)  | **43.3** (37.2)  |
| landbird class  | **87.9** (71.0)  | **98.0** (96.4)  |

Table 13: **Zero-shot classification accuracy on Waterbirds dataset, per class and background (ViT-H).** The accuracy for the baseline CLIP model is in parentheses.

|  | water background | land background |
|---|---|---|
| waterbird class | 80.5 (**86.1**) | **81.6** (63.5) |
| landbird class | **57.5** (45.6) | 94.3 (**96.1**) |

Table 14: **Zero-shot classification accuracy on Waterbirds dataset, per class and background (ViT-B)**. The accuracy for the baseline CLIP model is in parentheses.

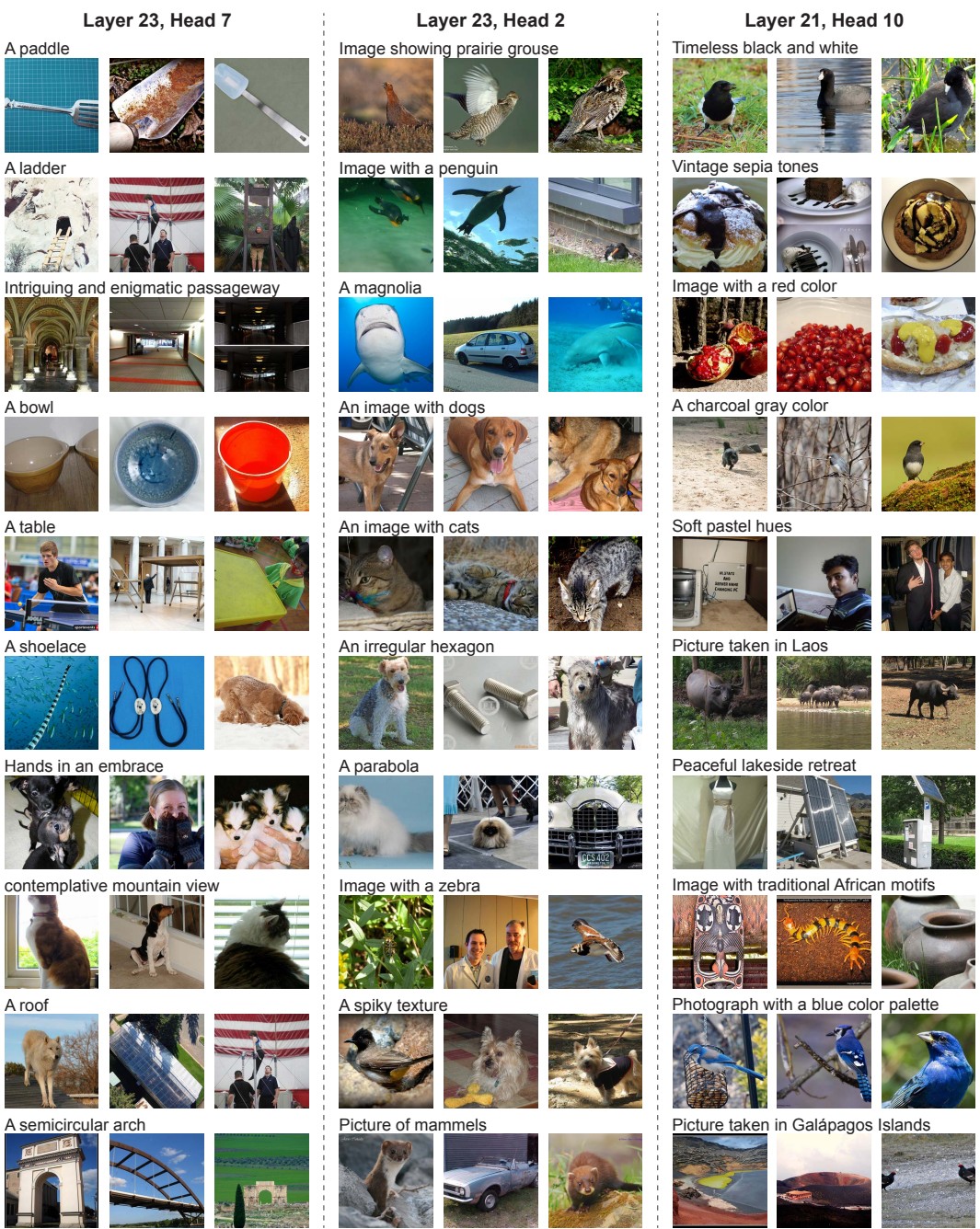

Figure 11: **Top 3 images with highest similarities to TEXTSPAN outputs.** For 3 randomly selected attention heads, we retrieve the images with the highest similarity score between their head contributions $c_{head}^{l,h}$ and the top 10 text representations found by our algorithm.

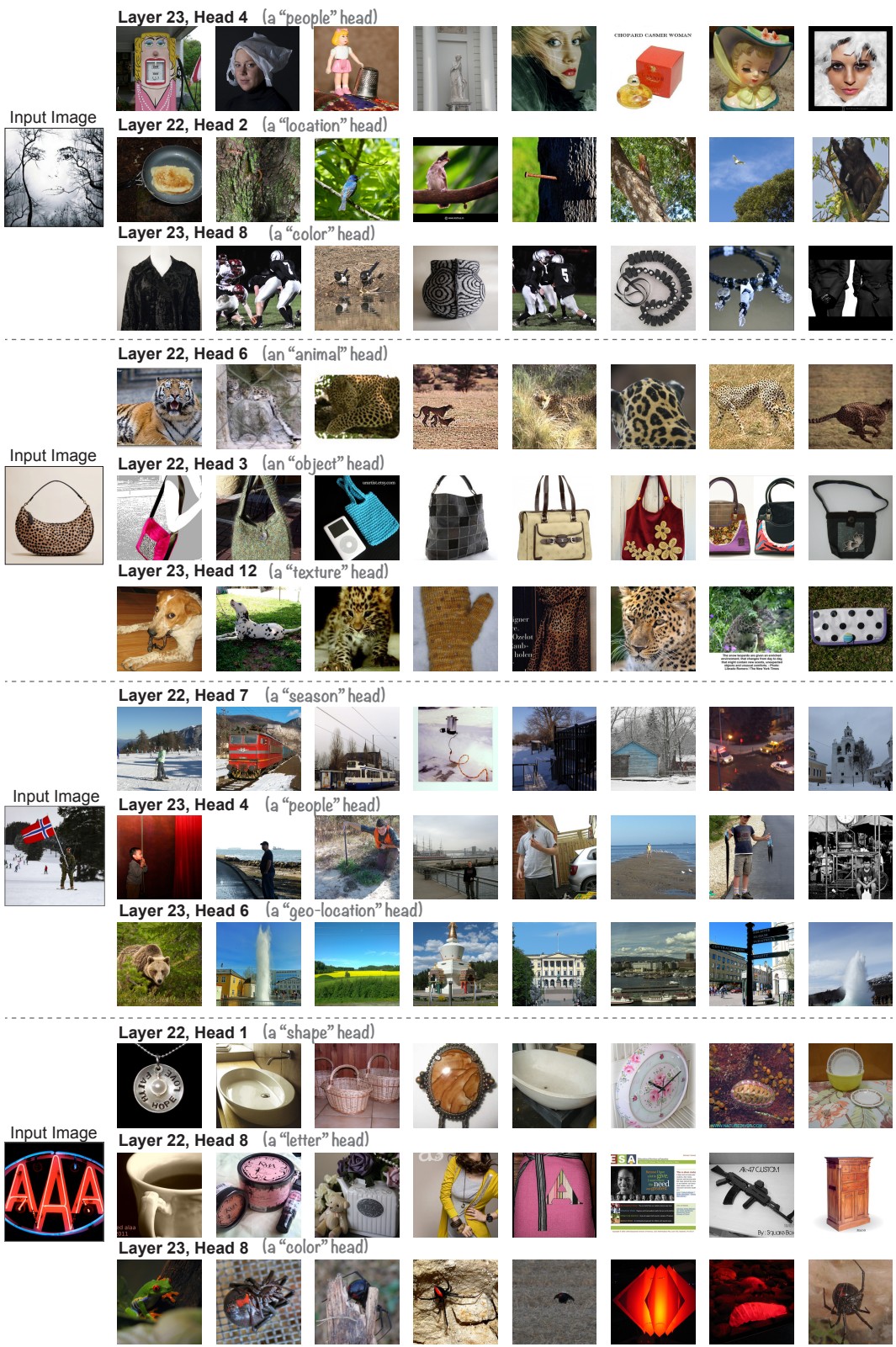

Figure 12: **Additional results for image retrieval based on head-specific similarity.**

