# OpenReview forum: "Interpreting CLIP's Image Representation via Text-Based Decomposition"
_ICLR.cc/2024/Conference — ICLR 2024 oral_

### Official Review · Reviewer_bDCR · 2023-10-28

**Soundness:** 4 excellent
**Presentation:** 4 excellent
**Contribution:** 4 excellent
**Rating:** 8
**Confidence:** 4

**Summary:**

The research dissects the CLIP image encoder, identifying specialized roles of its internal components, and reveals an inherent spatial awareness in its processing by using the CLIP’s text representation. Insights gained from this analysis facilitate the removal of extraneous data and the enhancement of the model, notably demonstrated through the creation of an effective zero-shot image segmenter. This underscores the potential for in-depth, scalable understanding and improvement of transformer models.

**Strengths:**

1. Well written text
2. Excellent figure to explain the pipeline. (Fig 1)
3. Tested on various backbone and datasets
4. Carefully designed ablation study to show why we should focus on MSA blocks instead MLP blocks. This also provides nice reasoning for the decomposition algorithm design.

5. An excellent way to provide explanations to researchers to understand the trained models. Instead of providing an enormous amount of random feature number to explain the model, the proposed method is able to align the reasoning to text. This could serve as a nice tool to collaborate with human researchers.
6. Smart way to utilize ChatGPT 3.5 to provide text prompts

**Weaknesses:**

1. Seems like human users are still required to provide some sort of heuristic to decide the role of a head. I would like to know how hard it is from the user point of view.
2. Seems like most of the experiments have been done on general image datasets. I am curious about the results on some fine grained tasks or datasets for example human face recognition.

**Questions:**

1. Human AI teaming.
- I am thinking about the task from a human-AI taming point of view. How hard is it for the human to identify the role of each head? Is it possible to introduce some humans during inference time to prune/select which head to use for final prediction?
2. “Note everything has a direct role”
- Does it mean there exists complicated features that we need to leave it as a black box, or we can ignore them as redundant feature points?

---

> ### Author Response · Authors · 2023-11-13
>
> We thank the reviewer for the valuable comments.
>
> __I am thinking about the task from a human-AI taming point of view. How hard is it for the human to identify the role of each head? Is it possible to introduce some humans during inference time to prune/select which head to use for final prediction?__
>
> In our experiments, we manually annotated the heads by examining the output text descriptions of our algorithm. For many of the heads, this task is simple (requires finding a commonality between 60 text descriptions). A possible approach for automating it is to query an LLM to summarize the role of the head based on its text descriptions (e.g. “What is common between all these image descriptions?”). Moreover, for pruning the heads for a specific task (e.g. bird classification), we can provide the discovered head roles to an LLM, and query it about each spurious ques that each role can provide for the given task (e.g. “Can geo-location be a spurious cue for bird image classification?”). This model-driven strategy would probably be more efficient than introducing humans during inference time, which might be too costly.
>
> __“Note everything has a direct role” - Does it mean there exists complicated features that we need to leave it as a black box, or we can ignore them as redundant feature points?__
>
> By “not everything has a direct role”, we are referring to the fact that our analysis studies the direct effects of individual components on the output representation rather than indirect effects.
>
> In more detail, the direct effects are the contributions of each layer to the residual stream (which is later projected into the output space). The indirect effects are the contributions of a layer that are processed by subsequent layers. Most of the early layers, for example, have meaningful indirect effects - they contribute to the inputs of later layers. To analyze these features, one should examine these contributions and their downstream effects. As a preliminary example of this, we unrolled the direct effect of a “counting head”, and found a second-order effect from an MLP layer that fires mostly when digits appear in the image, which could produce spurious cues if the presented digit usually corresponds to the number of objects in the images (e.g. a pack of 6 tomatoes with the text “6 tomatoes” appearing on the pack). Removing the second-order effect of this MLP on this head may result in more accurate counting abilities.

---

> > ### Comment · Reviewer_bDCR · 2023-11-17
> >
> > Thanks for the clarification. I noticed my typo from my question. Sorry about that.
> >
> > I found your proposed method could be useful to human observers. Nice work.

---

### Official Review · Reviewer_eYDd · 2023-10-31

**Soundness:** 3 good
**Presentation:** 4 excellent
**Contribution:** 3 good
**Rating:** 8
**Confidence:** 4

**Summary:**

The paper ’Interpreting CLIPs Image Representation via Text-based Decomposition’ studies the internal representation of CLIP by decomposing the final [CLS] image representation as a sum across different layers, heads and tokens. Using direct effect to ablate certain layers, the authors first understand that the late attention layers are important for the image representation. Furthermore, the authors further study the attention heads and token decompositions in the later layers to come up with two applications: (i) Annotating attention heads with image-specific properties which was leveraged to primarily reduce spurious cues; (ii) Decomposition into image tokens enabled zero-shot segmentation.

**Strengths:**

- This paper is one of the well executed papers in understanding the internal representations of CLIP. The writing is excellent and clear!
- The text-decomposition based TextBase is technically simple, sound and a good tool to annotate internal components of CLIP with text. - This tool can infact be extended to study other non-CLIP vision-language models too.
- Strong (potential) applications in removing spurious correlation, image retrieval or zero-shot segmentation.

**Weaknesses:**

Cons / Questions:
- While the authors perform the direct effect in the paper, can the authors comment on how indirect effects can be leveraged to understand the internal representations? I think this is an important distinction to understand if understanding the internal representations in more detail can unlock further downstream capabilities. If affirmative, what downstream capabilities will be feasible?
- I am not sure if the current way to create the initial set of descriptions is diverse enough to capture “general” image properties or attributes. I believe the corpus of 3.4k sentences is too small for this analysis. While this set is a good starting point, can the authors comment on how this set can be extended to make it more diverse?
- Did the authors analyse the OpenAI CLIP variants using this framework? The OpenCLIP and OpenAI variants are trained on different pre-training corpus, so a good ablation is to understand if these properties are somehow dependent on the pre-training data distribution.
- Can the authors comment on how the images set in Sec. 4.1 is chosen? This is not very clear from the text. Is this set a generic image set that you use to obtain m text-descriptions per head from a bigger set of text-descriptions?

**Questions:**

Check Cons / Questions;

Overall, the paper is an interesting take on understanding the internal representations of CLIP with the additional benefit of showing applications on image retrieval and reducing spurious correlations.  I am leaning towards acceptance and happy to increase my score if the authors adequately respond to the questions.

---

> ### Author Response · Authors · 2023-11-13
>
> We thank the reviewer for the valuable comments.
>
> __"can the authors comment on how indirect effects can be leveraged to understand the internal representations?"__
>
> Examining the indirect effect can result in a finer understanding of the internal computation graph in these models. For example, we can analyze the second-order effects going through a specific layer/head (e.g. what does MLP 2 contribute to attention head 5 in layer 30).
> This way, one can remove more fine-grained spurious correlations.
>
> As a preliminary example of this, we unrolled the direct effect of a “counting head”, and found a second-order effect from an MLP layer that fires mostly when digits appear in the image, which could produce spurious cues if the presented digit usually corresponds to the number of objects in the images (e.g. a pack of 6 tomatoes with the text “6 tomatoes” appearing on the pack). Removing the second-order effect of this MLP on this head may result in more accurate counting abilities.
>
> __“Did the authors analyse the OpenAI CLIP variants using this framework? The OpenCLIP and OpenAI variants are trained on different pre-training corpus, so a good ablation is to understand if these properties are somehow dependent on the pre-training data distribution.”__
>
> We provide here an additional comparison between OpenCLIP-ViT-L14 and OpenAI’s CLIP-ViT-L14. First, our observation that the last few attention layers have most of the direct effects on the final representation still holds: keeping the last 5 attentions, and ignoring the direct effects of other layers, result in only a small decrease from 75.5% to 73.2% in accuracy on ImageNet.
> Second, we evaluate OpenAI's CLIP-ViT-L-14 for zero-shot image segmentation:
>
> | model | mIoU | Pixel-wise Acc. | MAP |
> |---------------------|-----------|-----------|-----------|
> | OpenAI ViT-L-14 | 55.24 | 76.19 | 81.37 |
> | OpenCLIP ViT-L-14 | 54.50 | 75.21 | 81.61 |
>
> As shown here, the localization properties of OpenAI-CLIP are comparable to the results of OpenCLIP.
>
> __“While this (initial descriptions pool) set is a good starting point, can the authors comment on how this set can be extended to make it more diverse?”__
>
> We repeat our experiments with a larger and more diverse image description corpus and present the results in section A.4 in the revised version. We query ChatGPT for class-specific descriptions and create an additional pool of 28767 unique sentences (8 times larger than our previous corpus). This results in higher accuracy with fewer basis vectors per head (smaller $m$) compared to the other pools that contain 3498 vectors, as shown in Figure 7.
>
> Aside from this, and more importantly for future work, our method can be made adaptive, by applying it iteratively and using a data-driven refinement step to update the pool. Specifically, at each iteration, we can apply our algorithm to retrieve a basis set and the corresponding descriptions (which indicates roughly what concepts the model is using), query an LLM to generate “more like these” descriptions (to retrieve more fine-grained descriptions), and add the generated new examples to the pool before re-computing the basis set again. This approach may result in more fine-grained descriptions that capture better the output space of each head. We plan to explore this idea further in future work.
>
> __Clarification about the image set in Sec. 4.1__
>
> We apply our algorithm on ImageNet validation set (50000 images), as it is the largest and most diverse dataset we could use given our compute budget.

---

> > ### Comment · Reviewer_eYDd · 2023-11-20
> > **Response to Authors**
> >
> > I thank the reviewer for their responses. They have resolved my comments and therefore I increase my rating as promised!

---

### Official Review · Reviewer_uT2r · 2023-11-01

**Soundness:** 3 good
**Presentation:** 3 good
**Contribution:** 3 good
**Rating:** 8
**Confidence:** 2

**Summary:**

The paper delves into the analysis of the CLIP image encoder, breaking down the image representation by considering individual image patches, model layers, and attention heads. Using CLIP's text representation, the authors interpret these components, discovering specific roles for many attention heads, such as identifying location or shape. They also identify an emergent spatial localization within CLIP by interpreting the image patches. Leveraging this understanding, they enhance CLIP by eliminating unnecessary features and developing a potent zero-shot image segmenter. The research showcases that a comprehensive understanding of transformer models can lead to their improvement and rectification. Furthermore, the authors demonstrate two applications: reducing spurious correlations in datasets and using head representations for image retrieval based on properties like color and texture.

**Strengths:**

Strength:
1. Paper is well-organized
2. The proposed analysis (importance of different attention layers) and two use cases (removal of spurious correlations and head-based image retrieval) are interesting.

**Weaknesses:**

Weakness:
No ablation studies on the impact of the pool size (M) and the basis size (m) on the performance of the decomposition.

**Questions:**

Questions:
1. “all layers but the last 4 attention layers has only a small effect on CLIP’s zero-shot classification accuracy” maybe just because the early layers’ feature are not semantically distinctive? But they should be still important to extract low-level features.
2. Is it possible to achieve the “dual” aspect of the text encoder of the CLIP: 1) find layer-wise importance of the text encoder; 2) find and remove redundant heads to reduce spurious correlations; 3) perform head-based text retrieval based on query images.

---

> ### Author Response · Authors · 2023-11-13
>
> We thank the reviewer for the valuable comments.
>
> __“No ablation studies on the impact of the … basis size (m)”__
>
> We believe that Figure 3 already implicitly contains the desired ablation. Specifically, in Figure 3 we simultaneously replace each head’s direct contribution with its projection to the $m$ text representations found by our algorithm. We consider a variety of output sizes $m \in \\{10, 20, 30, 40, 50, 60\\}$. We evaluate the reconstruction by comparing the downstream ImageNet classification accuracy. As shown in the figure and discussed in section 4.2 the accuracy improves with larger $m$ values. We found that 60 descriptions ($m=60$) were sufficient to reach an accuracy that is close to the original model accuracy, but with fewer descriptions, the accuracy drops.
>
> __“No ablation studies on the impact of the pool size (M)”__
>
> To vary the pool size $M$, we repeat our experiments with a larger and more diverse image description corpus and present the results in section A.4 in the revised version. We query ChatGPT for class-specific descriptions and create an additional pool of $M=28767$ unique sentences (8 times larger than our previous corpus). This results in higher accuracy with fewer basis vectors per head (smaller $m$) compared to the other pools that contain $M=3498$ vectors, as shown in Figure 7 in the revised version.
>
> __“all layers but the last 4 attention layers has only a small effect on CLIP’s zero-shot classification accuracy” maybe just because the early layers’ feature are not semantically distinctive? But they should be still important to extract low-level features.__
>
> In our analysis, we consider only the direct effects and ignore the indirect effects (which would include extracting low-level features that get used in later stages of processing). Our claim is only that the direct effect of the early layers on the output is negligible (Figure 2). In contrast, we expect the indirect effects to be large, since the early layers contribute to the inputs of later layers.
>
> __“Is it possible to achieve the “dual” aspect of the text encoder of the CLIP”__
>
> While we think that decomposing the text encoder is out-of-scope for this paper, we believe that such a dual analysis is possible, since the text encoder has a similar transformer architecture. Moreover, decomposing both the text encoder representation and the image encoder representation will allow us to examine what each (patch, word) pair contributes to the overall similarity score. We plan to explore it in future work.

---

> > ### Author Response · Authors · 2023-11-21
> >
> > Dear reviewer,
> >
> > Towards the end of the discussion phase, we trust that our response has successfully addressed your inquiries. We look forward to receiving your feedback regarding whether our reply sufficiently resolves any concerns you may have, or if further clarification is needed.
> >
> > Thank you,
> >
> > Authors

---

### Official Review · Reviewer_ETim · 2023-11-01

**Soundness:** 3 good
**Presentation:** 4 excellent
**Contribution:** 3 good
**Rating:** 8
**Confidence:** 4

**Summary:**

This paper delves deep into the famous CLIP vision-langauge model for understanding its image-encoder components with the help of text representation. This includes understanding the role of attention heads, self-attention layers, MLP layers as well as the effect of each layer on the final representation in terms of downstream task performance. As the image backbone is completely aligned with text embedding representations, this work makes advantage of this property to interpret the image encoder components.
Interestingly, each head role is associated with capturing specific attributes and properties, which leads to several downstream applications including, image retrieval with specific properties defined by head and removing spurious features by neglecting the heads which focuses such features.
The analysis is performed on CLIP models with various scales, which shows the generality of this study.

**Strengths:**

1) This paper presents important timely study to understand the inner working of image backbones of the large scale vision-language models like CLIP which has become a very popular paradigm. This will help to improve the next generation of such models in terms of architecture and training.
2) The use of text to interpret the image backbone components is motivating by the fact that CLIP backbone understands text representations.
3) Extensive analysis shows the main performance drivers of the image backbone of CLIP, and could help in rejecting the redundant modules present in such vision-language architectures.
4) The proposed TextBase algorithm to associate specific roles per head using text is fairly motivated and its effectiveness is shown via downstream retrieval experiments.
5) This analysis unlocks several improvements for downstream tasks including reducing known spurious cues and zero-shot segmentation.
5) Finally the paper is well written and easy to understand.

**Weaknesses:**

I could not think much about any significant weaknesses in this work. I have some questions as follows:

1) How the zero-shot results compare against from methods like MaskCLIP [1]?
2) It has been shown that the zero-shot accuracy of the embeddings from only late attention layers is very competitive to the original performance. Will this also hold true where the representations are used for downstream tasks which require adaption? For example, it will be good to see the linear probe performance of the filtered embeddings on imagenet or other datasets like CIFAR100, Caltech101.







[1] Extract Free Dense Labels from CLIP, ECCV 2022, Oral

**Questions:**

Please refer to the weaknesses section! Thank you.

---

> ### Author Response · Authors · 2023-11-13
>
> We thank the reviewer for the valuable comments.
>
> __“the zero-shot accuracy of the embeddings from only late attention layers is very competitive with the original performance. Will this also hold true where the representations are used for downstream tasks that require adaption?”__
>
> To verify that the effect of the early attention layers and MLPs is negligible even after adaptation, we followed the suggested experiment and applied linear probing on ViT-B-14 for CIFAR100 classification. The test set accuracy of linear probing is 84.3. When we mean-ablate all the MLPs and the early attention layer up to the last 4 layers we get an accuracy of 84.2%. This suggests that even after adaptation, the late attention layers still have most of the effect on the output.
>
> __“How do the zero-shot results compare against methods like MaskCLIP”__
>
> MaskCLIP uses a different approach, that works only on CLIP models that have attention pooling before the projection layer. Differently from these models, other CLIPs (and specifically - all the OpenCLIP models) use the class token directly as the input to the final projection, without using attention pooling. We note that it is possible to compare our model to maskCLIP by modifying the decomposition to include attention pooling in it, but we think it is out of the scope of this paper.

---

> > ### Comment · Reviewer_ETim · 2023-11-18
> >
> > Thank you for providing a response to my queries!
> >
> > The results of Linear probing with just attention layers-based embeddings seem encouraging!
> >
> > Regarding the MaskCLIP, I actually was referring to comparing current zero-shot segmentation results (presented in your paper) and MaskCLIP results on the same datasets.

---

> > > ### Author Response · Authors · 2023-11-19
> > >
> > > Thank you for the clarification.
> > >
> > > We compared our decomposition to MaskCLIP for zero-shot segmentation. We used OpenAI's ViT-B-16 and calculated metrics on Imagenet Segmentation dataset:
> > >
> > > | model | mIoU | pixel-wise Acc. | MAP |
> > > |--------|------|-------|-------|
> > > |MaskCLIP| 54.8 | 73.0 | **83.6**|
> > > |Ours| **57.7** | **77.2** |82.6|
> > >
> > > As shown, our decomposition is better in pixel-wise accuracy and mIoU, and slightly worse in MAP.

---

> > > > ### Comment · Reviewer_ETim · 2023-11-20
> > > >
> > > > Thank you for providing the results and they look encouraging.
> > > >
> > > > Overall I am satisfied with the paper and will keep my rating as it is.

---

### Meta-Review · Area_Chair_kvw2 · 2023-12-11

**Metareview:**

This paper provides an interesting and scalable interpretability study about CLIP.  Specifically, it leverages the alignment of the image backbone with text representations, allowing for insightful interpretations of each head's role in capturing specific attributes. This interpretability also enables several downstream applications, including targeted image retrieval and effective zero-shot image segmenter. Overall, all reviewers enjoy reading this paper, and highly appreciated its in-depth interpretability analysis of CLIP, with interesting applications. There are only a few minor concerns raised, mainly about further ablations/clarifications to improve the quality of this work. The rebuttal addresses all of them; all reviewers unanimously (and strongly) recommend accepting it.

**Justification For Why Not Higher Score:**

N/A

**Justification For Why Not Lower Score:**

I (personally) believe this paper brings a very novel approach to (scalably) interpreting the CLIP model. By leveraging the alignment between image backbones and text representations, the paper not only deepens our understanding of CLIP but also demonstrates practical and interesting applications. Furthermore, the strong endorsement from reviewers, emphasizing the paper's in-depth analysis and practical implications, underscores its high quality and relevance.

---

### Decision · Program_Chairs · 2024-01-16

Accept (oral)